# Isolation of infectious Lloviu virus from Schreiber's bats in Hungary

Gábor Kemenesi [1,2,11 ✉], Gábor E. Tóth [1,2,11], Martin Mayora-Neto [3], Simon Scott [3], Nigel Temperton [3], Edward Wright[4], Elke Mühlberger[5], Adam J. Hume [5], Ellen L. Suder [5], Brigitta Zana[1], Sándor A. Boldogh [6], Tamás Görföl [1], Péter Estók [7], Tamara Szentiványi[8], Zsófia Lanszki[1,2], Balázs A. Somogyi[1], Ágnes Nagy[9], Csaba I. Pereszlényi[9], Gábor Dudás[9], Fanni Földes[1], Kornélia Kurucz[1,2], Mónika Madai[1], Safia Zeghbib[1], Piet Maes [10], Bert Vanmechelen [10] & Ferenc Jakab[1,2]

Some filoviruses can be transmitted to humans by zoonotic spillover events from their natural host and filovirus outbreaks have occured with increasing frequency in the last years. The filovirus Lloviu virus (LLOV), was identified in 2002 in Schreiber's bats (*Miniopterus schreibersii*) in Spain and was subsequently detected in bats in Hungary. Here we isolate infectious LLOV from the blood of a live sampled Schreiber's bat in Hungary. The isolate is subsequently sequenced and cultured in the *Miniopterus* sp. kidney cell line SuBK12-08. It is furthermore able to infect monkey and human cells, suggesting that LLOV might have spillover potential. A multi-year surveillance of LLOV in bats in Hungary detects LLOV RNA in both deceased and live animals as well as in coupled ectoparasites from the families Nycteribiidae and Ixodidae. This correlates with LLOV seropositivity in sampled Schreiber's bats. Our data support the role of bats, specifically *Miniopterus schreibersii* as hosts for LLOV in Europe. We suggest that bat-associated parasites might play a role in the natural ecology of filoviruses in temperate climate regions compared to filoviruses in the tropics.

[1] National Laboratory of Virology, Szentágothai Research Centre, University of Pécs, Pécs, Hungary. [2] Institute of Biology, Faculty of Sciences, University of Pécs, Pécs, Hungary. [3] Viral Pseudotype Unit, Medway School of Pharmacy, Chatham Maritime, Universities of Kent & Greenwich, Kent, UK. [4] Viral Pseudotype Unit, School of Life Sciences, University of Sussex, Falmer, Sussex, UK. [5] Department of Microbiology, Boston University School of Medicine, Boston, MA, USA. [6] Aggtelek National Park Directorate, Jósvafő, Hungary. [7] Department of Zoology, Eszterházy Károly University, Eger, Hungary. [8] Institute of Ecology and Botany, ÖK Centre for Ecological Research, Vácrátót, Hungary. [9] Medical Centre, Hungarian Defence Forces, Budapest, Hungary. [10] Leuven, Rega Institute, Department of Microbiology, Immunology and Transplantation, Laboratory of Clinical and Epidemiological Virology, Leuven, Belgium. [11] These authors contributed equally: Gábor Kemenesi, Gábor Endre Tóth. ✉email: kemenesi.gabor@gmail.com

Some members of the *Filoviridae* family are known to cause severe disease and frequent death in humans (e.g., ebolaviruses and marburgviruses), while there are other members with no known pathogenicity to humans (e.g., Reston virus)[1]. Among filoviruses, ebolaviruses have received the most attention, both by public health experts and the public, mainly due to the multiple documented human outbreaks, particularly the West African Ebola virus (EBOV) disease outbreak in 2013–2016 and the recent outbreaks in the Democratic Republic of the Congo and Guinea[2–4]. The first direct evidence pointing towards bats as natural reservoirs for EBOV was published in 2005, when RNA and virus-specific antibodies were detected in three species of fruit bats captured in Gabon and the Democratic Republic of the Congo[5]. To date, direct isolation of the infectious virus from a specific bat species has only been achieved in the case of Marburg virus (MARV), which leaves several open questions regarding the natural reservoir host for other filoviruses[6]. During the past few years, increased scientific focus on these viruses has revealed the presence of several novel filoviruses in bats from Asia[7–10], Africa[11–13] and Europe[14,15]. Currently, the only filovirus known to be endemic in Europe is Lloviu virus (LLOV), where genomic RNA was identified during the investigation of Schreiber's bats (*Miniopterus schreibersii*) die-off events in the Iberian-peninsula in 2002[14]. After its initial discovery, no other reports were published with regards to the presence of this virus until 2016, when RT-PCR positive bat carcasses in Hungary confirmed a more widespread presence of LLOV in *M. schreibersii* across Europe[15]. This was followed by a report on seropositivity among Schreiber's bats in Spain, suggesting the circulation of LLOV in that area[16]. The circumstances of the initial detection of LLOV in Hungary were similar to the original events in Spain, with multiple possibly related die-off events in the area, exclusively affecting Schreiber's bats[15]. Nearly two decades have now passed since the discovery of LLOV and many questions remain unresolved regarding the nature of the virus, the risk of zoonotic spillover, and most importantly, the pathogenic potential for bats and humans. Concerns regarding the possible pathogenicity of LLOV in humans remain high, since multiple studies have revealed its functional and genomic relatedness to EBOV[17–20]. Recently, a recombinant LLOV (rLLOV) rescue system was developed, in which the missing LLOV genome termini were complemented by homologous EBOV or MARV sequences. Although rLLOV is able to infect known target cells of EBOV, including human primary macrophages, the inflammatory response in human macrophages, a hallmark of Ebola virus disease, is not induced by rLLOV. This suggests that LLOV might be able to infect humans, but the infection might not lead to disease, similar to the nonpathogenic Reston virus[21]. Public health preparedness to filovirus zoonoses is highly related to the ecological attributes of wildlife hosts. Outbreaks in the human population are most likely initiated by spillover events from infected animals to humans[22]. In the case of MARV, these spillover events have been reported to correlate with the life cycle of the bat host, with increased zoonotic events occurring during the birthing period[23,24]. Importantly however, bat life cycles are different in temperate climate conditions than in the tropics. The annual life cycle of Schreiber's bats has key differences from bats that live in the tropics, most importantly the period of hibernation (Supplementary Fig. 1).

In the past two decades, the identification of reservoir species of filoviruses has become a major focus of research, with numerous different vertebrate and arthropod taxa being studied[25–27]. Interestingly, ectoparasites of bats have not been involved in many of these investigations, although they can carry several types of pathogens[28]. Bat flies are the most common ectoparasites of bats[29]. In the case of hippoboscoid flies, both sexes feed on the blood of the host species. Members of the Nycteribiidae and Streblidae families are typically associated with specific host species and in general bat flies and bats show a strong coevolution[30]. While bats are a major focus in viral emerging infectious disease research, the transmission patterns of these viruses within host bat populations and the role of their highly-specialized ectoparasites in this enzootic ecology have barely been investigated to date[28]. Ectoparasites, however, represent perfect vector candidates for bat-to-bat transmission, including intraspecies and interspecies transmission[31]. It is therefore important to understand the complex picture of LLOV transmission within bats and potential vector species, including ectoparasites, to get a clearer picture about the natural circulation of this, and other filoviruses.

In the present study, we sought to understand the biology of LLOV at a specific bat roost site in northeastern Hungary. We investigated the role of Schreiber's bats in the natural circulation of LLOV and whether bat-associated parasites might play a role in the ecology of LLOV. We performed serologic, RT-PCR and sequencing-based surveillance on Schreiber's bats from a site of previous LLOV detection in Hungary[15] following a step-by-step investigation strategy. A primary result was the isolation of infectious virus directly from a bat blood sample, which makes LLOV only the third member of the family *Filoviridae* ever isolated from bats and the first filovirus isolation of a non-Marburg or Ebola virus genus member.

We developed and used a deployable field-sequencing technology based on target enrichment method. It is not without precedent, as similar techniques were used extensively to sequence viruses from patient samples during the West African EBOV outbreak[32,33]. Here, we publish the use of such technology directly on wildlife hosts, optimized for the Nanopore sequencing platform that is amenable for both field and laboratory use in future studies.

## Results

**LLOV seropositivity in Schreiber's bats**. Blood samples were collected from live bats and bat carcasses during the period of 2016–2020 at the site of LLOV detection in Northeastern Hungary in 2016[15]. Live bats were apparently healthy, whilst dead bats were found in various conditions, possibly depending on the incubation period of the carcass within cave conditions. All details about the samples and animals, such as sex and collection date are listed as Supplementary Data 1. In order to perform neutralization tests to detect the presence of anti-LLOV antibodies, we created lentivirus particles pseudotyped with the LLOV or EBOV glycoprotein (GP). LLOV PVs did not cross react with EBOV convalescent serum (NIBSC WHO standard 15/262) (Supplementary Fig. 4). The converse neutralizing cross-reactivity was tested using bat sera and an EBOV PV target. However, due to the insufficient remaining volumes of bat sera only two LLOV nAb positive bat sera and two LLOV nAb negative bat sera could be tested, in addition to EBOV convalescent serum as a positive control. Importantly, no cross-reactivity was observed in these experiments (Supplementary Fig. 4). These findings are in accord with the lack of inter-genera filovirus cross-reactivity previously reported between Ebola and Marburg viruses using an indirect ELISA[34].

Following the cross-reactivity tests with EBOV, we used the LLOV PVs to perform antibody neutralization tests (PVNTs) on bat serum samples from both live and dead bats (Fig. 1, Table 1). Nine Schreiber's bat carcasses were collected during the period between 2016 and 2019 and from these four tested seropositive for LLOV, with relatively high neutralization titres (254–3485 $IC_{50}$) (Supplementary Data 1). We were also able to detect LLOV RNA in certain tissues of these carcasses with low copy numbers,

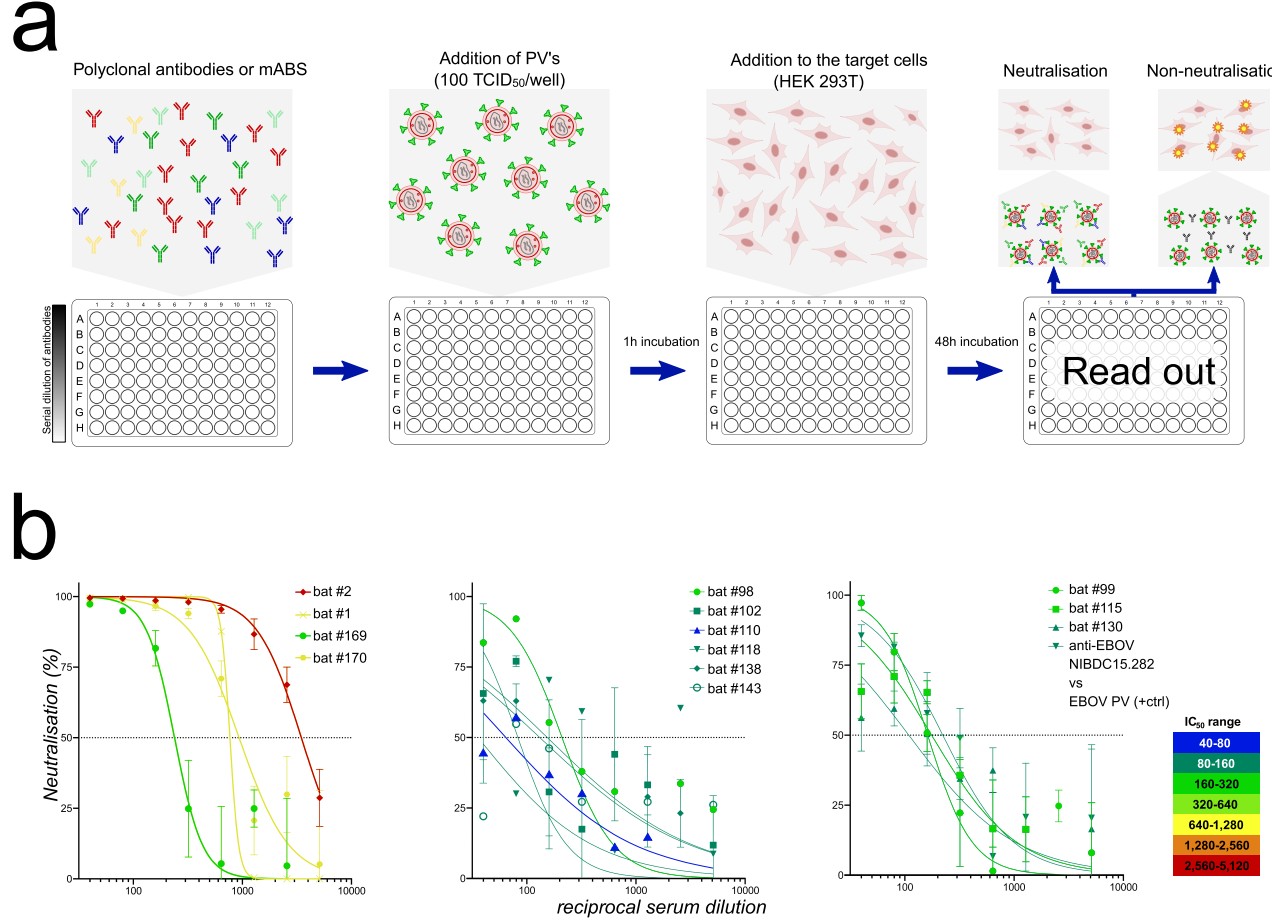

**Fig. 1 Pseudotyped virus neutralization test (PVNT) experiments of blood sera from Schreiber's bats. a** Workflow of PVNT; **b** PVNT results of LLOV seropositive bat sera. First graph represents dead animal samples (samples: bat #2, bat #1, bat #169, bat #170), whilst the second and third graph presents live bat samples (bat #98 to bat #130). Anti-EBOV NIBSC15.282 vs. EBOV PV data is also shown as a matched positive technical control. Error bars indicate mean ± SD. Source data are provided as a Source Data file.

**Table 1 PVNT results of seropositive bats paired with RT-PCR results.**

| Animal ID | PVNT LLOV 1 IC$_{50}$ | PVNT LLOV 2 IC$_{50}$ | Mean IC$_{50}$ | Serology | Real-time RT-PCR | Status | Collection date |
|---|---|---|---|---|---|---|---|
| LLOV 98 | 211 | a | 211 | Pos | N/A | Alive | 2018.09.18. |
| LLOV 99 | 156 | 174 | 165 | Pos | N/A | Alive | 2018.09.18. |
| LLOV 102 | 132 | a | 132 | Pos | Neg | Alive | 2018.09.18. |
| LLOV 110 | 64 | a | 64[b] | Pos | Neg | Alive | 2018.09.18. |
| LLOV 115 | 143 | 197 | 170 | Pos | N/A | Alive | 2018.09.18. |
| LLOV 118 | 155 | a | 155 | Pos | Neg | Alive | 2018.09.18. |
| LLOV 130 | 177 | 59 | 118 | Pos | Neg | Alive | 2018.09.18. |
| LLOV 138 | 83 | a | 83 | Pos | Neg | Alive | 2018.09.18. |
| LLOV 143 | 90 | a | 90 | Pos | Neg | Alive | 2018.09.18. |
| LLOV 1[c] | 768 | N/A | 768 | Pos | Pos (lung, spleen) | Dead | 2016.02.11. |
| LLOV 2[c] | 2999 | 3972 | 3485 | Pos | Neg | Dead | 2016.02.11. |
| LLOV 169[c] | 172 | 337 | 254 | Pos | Pos (lung, spleen) | Dead | 2019.01.31. |
| LLOV 170[c] | 757 | 1136 | 946 | Pos | Neg | Dead | 2019.01.31. |
| Control 1 | Non-neut. | Non-neut. | | Neg | Neg | Alive | 2019.08.08. |
| Control 2 | Non-neut. | Non-neut. | | Neg | Neg | Alive | 2019.08.08. |

[a]PVNT not performed due to lack of sufficient serum for a repeat experiment.
[b]Borderline positive value.
[c]Dead at the original site of virus emergence, Northeast Hungary, Zemplén Mt.; Control 1 (*Myotis myotis*) and Control 2 (*M. schreibersii*) were used as healthy negative control animals from a distant bat roost site in Southwestern Hungary.

detailed in Tables 1, 2. We also investigated serum samples taken from 74 live Schreiber's bats from the first sampling event in September 2018. Seroprevalence among live-sampled bats was 9/74 (12.16%) with relatively weak to moderate titres (64–211 IC$_{50}$),

particularly compared to seropositive carcasses (Fig. 1, Table 1). Unfortunately, the highly limited amount of blood samples from live animals precluded replicate testing in certain cases, also hindering the possibility of coupled serologic and RT-PCR

**Table 2 Detailed information about LLOV RNA-positive samples.**

| Sample ID | Species | Collection date | Tissue(s) | Cycle threshold (Ct) | Genomic copy numbers (copies/mL) |
|---|---|---|---|---|---|
| LLOV_1[a] | *Miniopterus schreibersii* | 11.02.2016 | Lung; spleen | 35.56; 35.30 | $1.4 \times 10^6$; $1.6 \times 10^6$ |
| LLOV_5[a] | *Miniopterus schreibersii* | 11.02.2016 | Lung | 35.09 | $1.8 \times 10^6$ |
| LLOV_105 | *Miniopterus schreibersii* | 18.09.2018 | Blood | 26.16 | $2.3 \times 10^8$ |
| LLOV_105_P2 | *Nycteribia schmidlii* (P) | 18.09.2018 | Whole specimen | 34.08 | $3.2 \times 10^6$ |
| LLOV_147 | *Miniopterus schreibersii* | 19.09.2018 | Blood | 25.39 | $3.6 \times 10^8$ |
| LLOV_147_P2 | *Nycteribia schmidlii* (P) | 19.09.2018 | Whole specimen | 34.61 | $2.4 \times 10^6$ |
| LLOV_169[a] | *Miniopterus schreibersii* | 31.01.2019 | Lung; spleen | 38.59; 36.31 | $2.7 \times 10^5$; $9.4 \times 10^5$ |
| LLOV_329[b] | *Miniopterus schreibersii* | 23.09.2019 | Blood | 30.27 | $2.5 \times 10^7$ |
| LLOV_329_P1 | *Ixodes simplex* (P) | 23.09.2019 | Whole specimen | 36.10 | $1.1 \times 10^6$ |
| LLOV_329_P2 | *Nycteribia schmidlii* (P) | 23.09.2019 | Whole specimen | 35.78 | $1.3 \times 10^6$ |
| LLOV_378[b, c] | *Miniopterus schreibersii* | 24.09.2019 | Blood | 30.10 | $2.8 \times 10^7$ |
| LLOV_378_P1[c] | *Nycteribia schmidlii* (P) | 24.09.2019 | Whole specimen | 34.53 | $2.5 \times 10^6$ |

(P) ectoparasite.
[a]Carcasses.
[b]Sample was used for in vitro isolation efforts.
[c]Samples were subjected to viral genomic sequencing.

examination. Therefore we were able to conduct both serological and RT-PCR examination in only one case (LLOV_105) where LLOV RNA was detected along with seronegativity (Supplementary Data 1).

**LLOV RNA in bats and related ectoparasites.** After the verification of seropositivity in bats, we focused on detection of LLOV RNA via RT-PCR in consecutive sampling events to facilitate viral genome sequencing and virus isolation efforts. Regular checking for dead animals during the hibernation period resulted in 10 bat carcasses (nine *Miniopterus schreibersii*, one *Myotis myotis*). After dissection of these carcasses, we tested blood and multiple organ samples (brain, liver, lung, spleen, and kidney) for the presence of LLOV and detected viral RNA in two (2/5) dead bats from 2016 and one from 2018 (Table 2). Overall, LLOV prevalence among *Miniopterus* carcasses was 33.3% (3/9). Notably, LLOV RNA was only detected in spleen and lung samples, with low genomic copy numbers in both organs comparing to live bat samples where we found one to three orders of magnitude higher amounts of viral genomic copies (Table 2).

Altogether, 779 samples (351 blood, 89 feces, 19 urine, 320 ectoparasites) were collected from live animals (351—*Miniopterus schreibersii*, 2—*Myotis myotis*) in seven sampling events and tested for the presence of LLOV RNA (Fig. 2a). We performed on-site RT-PCR analysis for the detection of viral RNA in blood samples from live animals, while other sample types were tested under laboratory conditions. On-site RT-PCR permitted the targeted examination and re-sampling of RT-PCR positive bats. Overall, 1.14% (4/351) of live-sampled animals were positive for LLOV by RT-PCR. Only the mid-to-late September sampling of bats in 2018 (LLOV_105, LLOV_147) and 2019 (LLOV_329, LLOV_378) provided positive blood samples, with a 4% RT-PCR positivity rate in 2018 and 2.8% in 2019 (Fig. 2). We did not detect LLOV RNA-positive animals in 2020, but only limited sampling was conducted due to COVID-19 pandemic-related restrictions. None of the RT-PCR positive animals showed signs of disease, and their general condition and activities appeared normal.

We collected feces samples from an RT-PCR positive bat (LLOV_378) to test the shedding characteristics of the virus and found no evidence of the presence of LLOV RNA in these samples (Fig. 2). All other feces ($n = 89$) and urine ($n = 19$) samples were retrieved from animals with RT-PCR negative blood and were found to be negative as well.

In order to get a clearer picture of possible bat-to-bat transmission pathways, we also examined the presence of LLOV RNA

in bat-related ectoparasites, mostly from the family Nycteribiidae known as "bat flies" along with some hard ticks (Ixodidae). In the process of collecting bat samples, specimens of bat flies including *Nycteribia schmidlii* ($n = 214$), *Nycteribia latreillii* ($n = 2$), *Penicillidia conspicua* ($n = 65$), and *Penicillidia dufouri* ($n = 4$), as well as *Ixodes simplex* ($n = 33$) and *Ixodes vespertilionis* ($n = 2$) ticks were collected. Positive ectoparasites were only retrieved from positive bats, suggesting the presence of the virus exclusively in engorged parasites (Fig. 2, Supplementary Data 1). Altogether, we detected LLOV RNA in four *Nycteribia schmidlii* samples and one *Ixodes* sample from four RT-PCR positive bats (Fig. 2b).

A detailed summary of all RT-PCR positive samples, including sample type/organ, $C_T$ values of RT-PCR reactions and indications of follow-up experiments are presented in Table 2.

**Lloviu virus isolation on bat, monkey, and human cells.** Freshly obtained blood samples from the animals previously denoted PCR positive (LLOV_329, LLOV_378) were used for in vitro isolation experiments. Considering the small body size of these bats, the volume of these samples is routinely extremely small, usually between 5 and 20 μl. The initial in vitro isolation efforts, utilizing multiple cell lines (Vero E6—African green monkey kidney and Tb1-Lu—bat lung cells) did not result in detectable LLOV replication. We neither observed any cytopathic effect in the cell culture experiments nor obtained positive real-time RT-PCR results after three blind passages.

After obtaining SuBK12-08 cells[35], we repeated these efforts using the only remaining LLOV_378 blood sample, since by this time we had used most of the positive sample aliquots in previous isolation efforts. SuBK12-08 is a *Miniopterus schreibersii*-originated, SV40 transformed kidney cell line. After the second blind passage, we observed a strong cytopathic effect (Fig. 3). Virus replication was further verified by real-time RT-PCR which resulted in lower real-time Ct value (18.469) than the original blood sample (30.1), clearly indicating an increase in viral genomic copy numbers. Viral titers of passage 4 virus were determined by tissue culture infectious dose ($TCID_{50}$) assay on SuBK12-08 cells and were as high as $1.78 \times 10^9$ $TCID_{50}$ units/ml. The established virus isolate was subjected for in vitro testing of other cell lines. Vero E6, SH-SY5Y (human neuroblastoma), Hep G2 (human hepatocyte carcinoma), HCC-78 (human lung adenocarcinoma), HCT 116 (human colon carcinoma) were successfully infected with MOI 0.01 virus isolate. After 10 days post-infection we measured a significant increase in viral genomic copy numbers. RT-PCR results of the cell supernatants are given

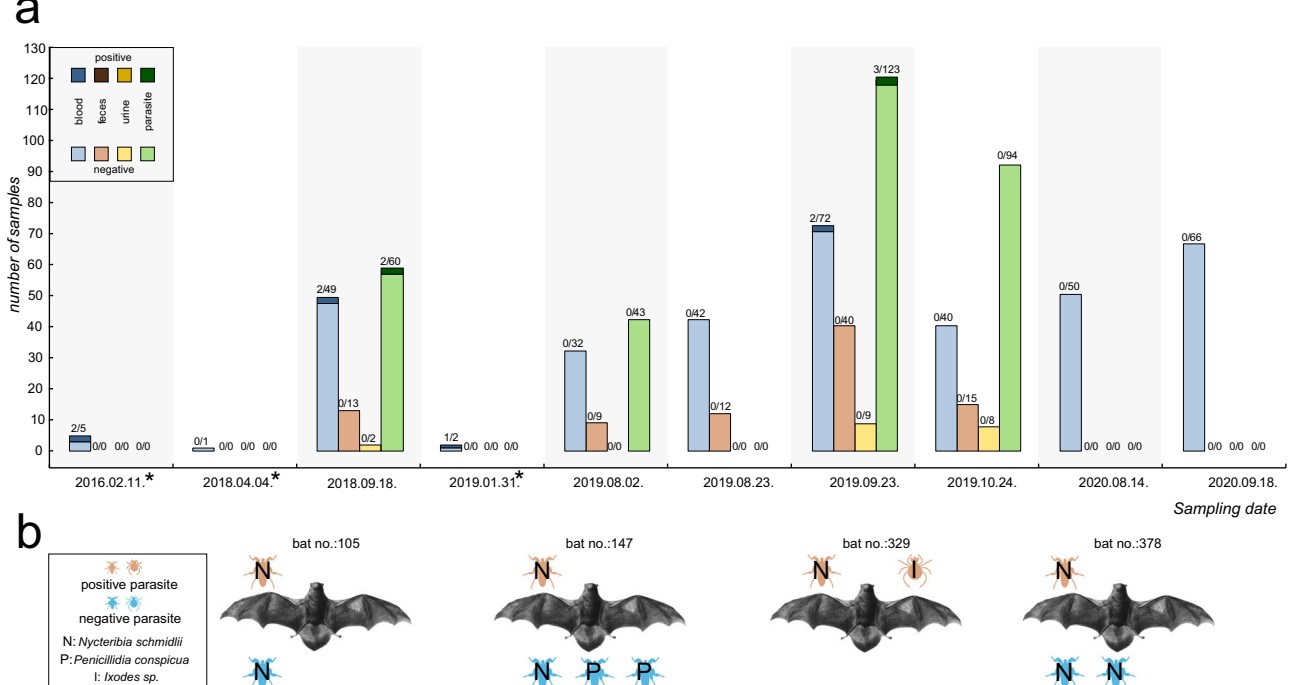

**Fig. 2 Summary of tested bat samples, verified RT-PCR positive bats and their ectoparasites. a** Number of LLOV-specific RT-PCR negative (lighter color) and positive (darker color) samples in each category: blood or organs (blue), feces (brown), urine (yellow), ectoparasites (green). *During these sampling events only passive monitoring was performed; (**b**) schematic representation of coupled bat-ectoparasite LLOV RT-PCR positivity.

(Supplementary Table 1). We present the visual progression of CPE on the SuBK12-08 (Supplementary Movie 1) and Vero E6 (Supplementary Movie 2) cell lines during LLOV infection.

Using the Vero E6 cell line, RNA fluorescent in situ hybridization (FISH) staining with LLOV genome-specific probes revealed a punctate pattern in LLOV-infected cells that was not observed in mock-infected control cells, confirming the presence of replicating LLOV in the infected cells (Fig. 3).

**Sequencing of the viral genome**. We developed an ARTIC-like amplicon-based (https://artic.network/) method for sequencing the complete coding region of the LLOV genome[32,36,37]. Using this method, we were able to retrieve sequence data from the RT-PCR positive bat from which virus was isolated (LLOV_378) and a nearly complete genome from its associated bat fly ectoparasite (LLOV_378_P1) (Table 2).

We generated sequence data for ~99% of the known LLOV reference genome (NCBI accession NC_016144). In the case of sample LLOV_378, 99.45% of the known genome was covered, starting from reference nucleotide position 28 and ending at 18,853. The mean coverage of this sample using all the data which was derived from the three different primer sets was 17,345× (Supplementary Fig. 2). The consensus sequence derived from the RT-PCR positive-associated parasite, sample LLOV_378_P1, spanned positions 28 to 18,853, with one notable gap between position 11,937 and 12,051. Importantly, there were no differences between the bat and bat fly derived viral sequences. Our LLOV sequence data are 99.198% identical to the reference genome. We detected the highest level of sequence variation between the sequence data and the reference genome within the glycoprotein gene 2 (GP2) (Fig. 4). We also performed an indicative sequencing of the established infectious isolate. The obtained genomic information for this isolate is available under the accession number (NCBI accession: MZ541881).

**Discussion**

Several novel members of the family *Filoviridae* have been discovered in the last decade including some found in Europe and Asia, raising the potential for the emergence of filoviruses outside of Africa. There is a considerable knowledge gap regarding the pathogenicity, animal hosts, and transmissibility of these newly discovered viruses.

The most groundbreaking result presented here is the successful isolation of infectious LLOV, from the blood sample of an RT-PCR positive bat. The pronounced cytopathic effect of LLOV-infected Miniopterus SuBK12-08 cells combined with the high titers of virus stocks propagated in these cells suggest that SuBK12-08 cells are highly permissive to LLOV infection. This, along with our other results, provides the first indication for a possible reservoir role of Schreiber's bats, which warrants further investigation to establish whether these bats can serve as asymptomatic hosts for the LLOV in Europe. The isolation of the virus opens the possibility for extended pathogenicity studies in the near future. As we also present here, Lloviu virus has the potential to infect human cells, therefore receptor identification, gene expression and indeed antiviral studies are now a priority.

We describe the circulation of LLOV in a colony of Schreiber's bats over the span of several years. This study represents the first detailed observation of the natural circulation of a filovirus in a temperate climate region. In addition to virus isolation results, the seropositivity and presence of LLOV RNA in these bats and the continuous circulation of the virus are supportive for the putative role of Schreiber' s bats as the natural reservoir hosts for this virus[38]. However at this point we can only conclude these animals as hosts for the virus in Europe. Our observations are partially in line with the previous finding that Egyptian fruit bats (*R. aegyptiacus*) are a reservoir for marburgviruses[39,40]. Other bat species have been reported as hosts for other filoviruses, such as the Bombali virus[11]. We detected LLOV RNA in multiple deceased bats, raising the possibility that these bats died as a

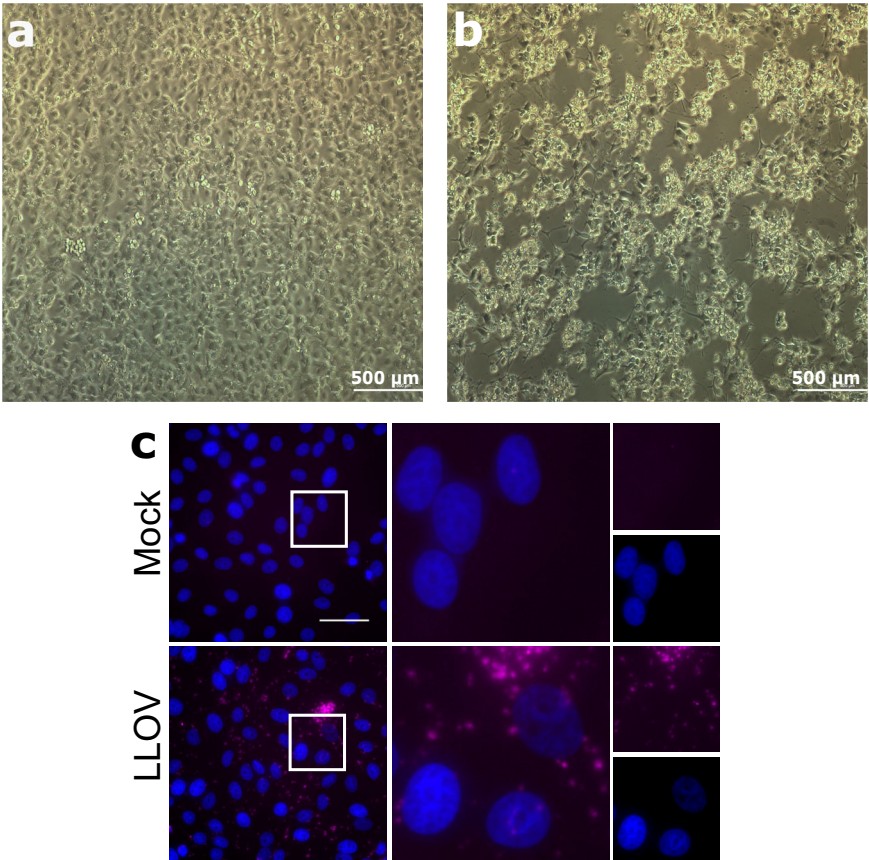

**Fig. 3 In vitro characterization of the Lloviu virus isolate.** Blind passage of Lloviu virus isolate and cytopathic effect on SuBK12-08 cells 10 days post infection (DPI), passage 4 (**a**, **b**) In situ hybridization of the Lloviu virus isolate on Vero E6 cells (**c**). **a** mock-infected cells 10 DPI, (**b**) Lloviu virus-infected cells 10 DPI, (**c**) Vero E6 cells were left uninfected (mock) or infected with LLOV at an MOI of 3. At 1 day post-infection, cells were fixed and stained by RNA FISH for viral genomic RNA (magenta). Cell nuclei were stained with DAPI (blue). Top row, mock-infected; bottom row, LLOV- infected. Inset square in left panels indicates region magnified in middle panels; far right panels display individual channels from middle panels. Scale bar represents 50 μm.

consequence of infection. Interestingly, some individuals were found with respiratory tract bleeding, however there was no correlation to RT-PCR positivity (Supplementary Table). Notably, the viral load in dead animals were one to three order of magnitude lower than in the live animal samples. More sophisticated surveillance activities are necessary to better understand the nature of LLOV infection in bats and to clarify the possible role of the virus in the morbidity and mortality among these bats. Although rare, the occurrence of both fatal and non-fatal outcomes of viral infections of bats is not unique. The best known pathogens to cause such patterns in bats are lyssaviruses[41].

We also developed a sensitive serological test based on pseudo-type virus neutralization that supports LLOV serosurveillance testing of bat sera samples. Before this study, only immunoblot analysis had been used for LLOV serologic examination in bats[16]. Our method enabled the serological study of live-sampled wild bats, showing a LLOV seroprevalence of 12.2% (9/74), which is lower than the previously published 36.5% seroprevalence from bats in Spain, though our test measures antibody neutralization rather than simply binding activity. This is also direct evidence of LLOV exposure in these bats, which supports the previous observation of a more extended spatio-temporal presence of the virus in Europe[15].

To exclude possible cross-reactivity with different filovirus nAbs we performed cross-reactivity tests (Supplementary Fig. 4). Based on the numerous European studies, involving viral meta-genomics or targeted surveillance there were no any indication for the presence of other filoviruses than LLOV in Europe so far[42]. In addition to these, during this study we performed a

secondary, pan-filovirus nested RT-PCR screening on our samples (Supplementary Data 1)[7]. We found no evidence for the circulation of other filoviruses than LLOV within the colony. Based on these data we do not expect the presence of other filoviruses which may interfere with the neutralization experiments. Also, we used GP-based PVNA system which was previously described as the most specific antigen compared to NP and VP40 antigens of filoviruses[43,44].

Except for MARV in Egyptian fruit bats, there is little information about enzootic circulation of a filovirus among specific bat colony sites. MARV virus prevalence values seen in various countries range from 2.5% and 13.7%, with seasonal variability noted[6,23,45]. Based on previous studies, natural fluctuation of the age composition within a single colony and therefore the changing numbers of susceptible animals within the population may largely affect the prevalence[23]. In case of Pteropodidae (fruit bats), bi-annual birth pulses may facilitate the persistence of filoviruses[24]. Available data on MARV circulation highlighted juvenile bats (around ~6 months old) as particularly likely to be infected at specific times of the year. This seasonal emergence of juvenile Egyptian fruit bats coincides with increased risk of human infection[23]. In contrast to some tropical species, *M. schreibersii* bats have only one breeding season per year, and notably have a hibernation period. Considering these features, there should be other driving factors involved in LLOV circulation (Supplementary Fig. 1).

The hibernation may also alter the response to LLOV infection as reactivation of latent herpesvirus infection or prolonged

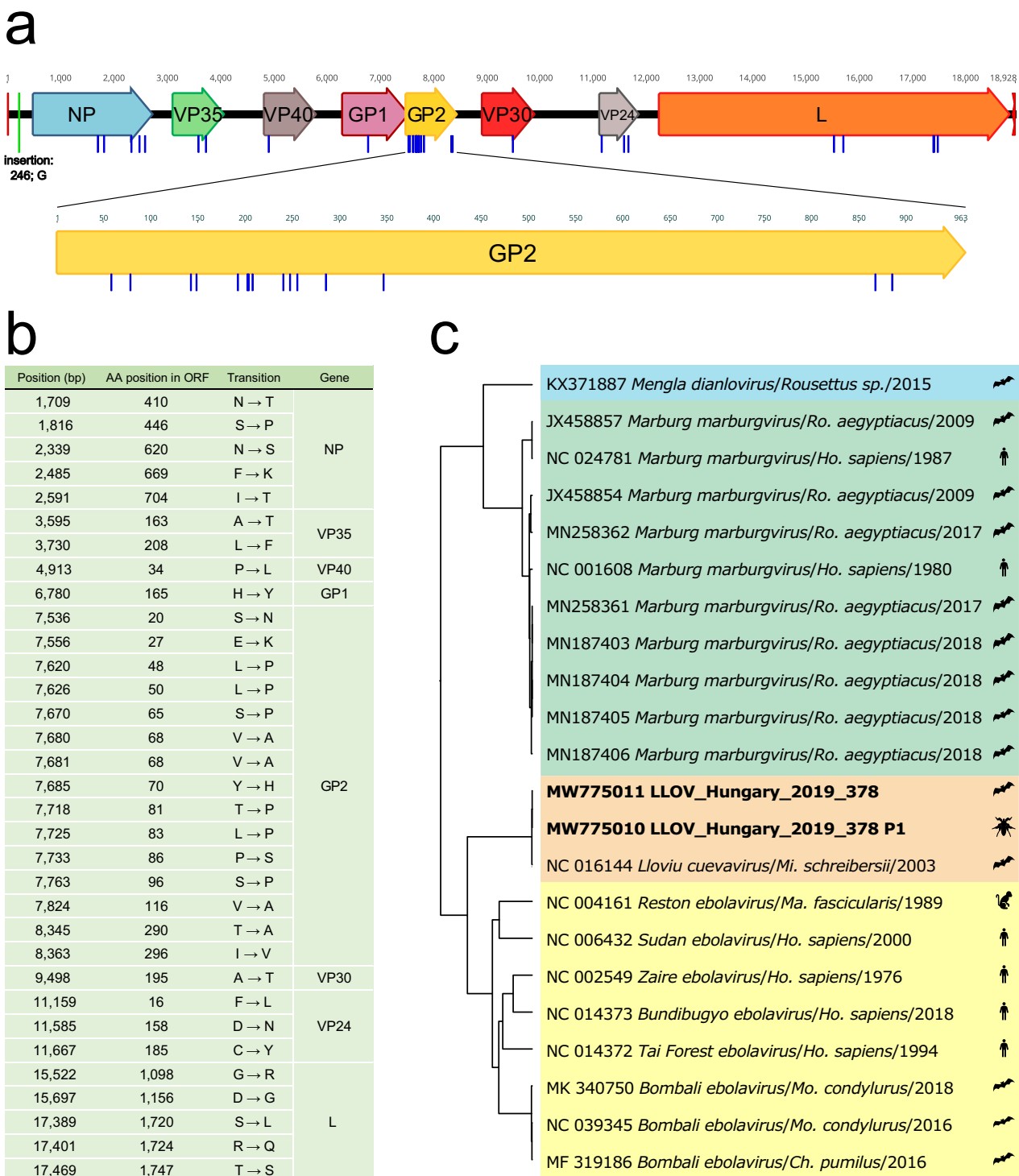

**Fig. 4 Genomic and phylogenetic attributes of the Lloviu virus genome sequences presented in this manuscript. a** Annotated Lloviu virus genome sequence from bat derived sample LLOV_378 (Genbank MW775011). Blue bars indicate differences in amino acid sequences and the green bar represents an insertion compared to the Spanish LLOV reference isolate (GenBank: NC_016144); (**b**) summary table of amino acid coding differences between LLOV_378 and NC_016144; (**c**) phylogenetic analysis of select complete genome sequences representing four filovirus genera including LLOV_378 (Genbank: MW775011) and LLOV_378P1 (Genbank: MW775010). Sequences are colored by genus *Dianlovirus* (blue), *Marburgvirus* (green), *Cuevavirus* (orange), and *Ebola virus* (yellow)). Host Taxon name abbreviations: Ro. *Rousettus*, Ho. *Homo,* Mi. *Miniopterus,* Ma. *Macaca,* Mo. *Mops,* Ch. *Chaerephon.*

coronavirus infection were both reported in bats in previous studies[46]. At this point it is hard to assess the causality of LLOV infection and fatal hibernation outcome in these bats, it is thus crucial to understand possible hibernation effects of LLOV

infection to better understand the pathogenic potential of this virus to these bats.

We have also provided evidence of filovirus RNA presence in arthropods. However, the precise role of bat flies and other

ectoparasites in LLOV transmission is unknown. It is not yet clear whether these bat flies may be acting as natural vectors or mechanical vectors for LLOV transmission or are simply dead-end LLOV spillover hosts. A recent article reported the unsuccessful infection of bat flies with MARV, ruling out the biological vector competence but still considered the possibility of mechanical transmission[47]. Our observations are in accordance with this recent report, since LLOV-positive ectoparasites were only retrieved from positive animals, suggesting the presence of the viral RNA in engorged parasites. Coupled sequencing results from host and parasite also support passive transmission without active viral replication, since no differences were found in the viral genomic sequences derived from the bat fly (LLOV_378_P1) and its bat host (LLOV_378). Given the current relatively limited dataset, it is not clear whether these ectoparasites play a role as a vector in the transmission of LLOV. However, we cannot rule out the possibility that these parasites serve as mechanical vectors in the transmission of LLOV, which warrants further studies. Compared to previous studies, Reston virus was not found to replicate in arthropod vectors and MARV RNA was not found in parasites on MARV-positive bats[48–50]. Based on the successful sequencing of LLOV RNA isolated from bat flies, resulting in high-quality sequence data, indirect LLOV surveillance could potentially be conducted by testing ectoparasites of *M. schreibersii* bats, obviating the need for invasive blood sampling of bats. This would provide the opportunity to conduct wide scale geographic surveillance to discover LLOV endemic regions and obtain a more detailed distribution map of LLOV infection among *M. schreibersii* populations. The reliability of ectoparasite-based surveillance strategy of bats was already reported for multiple bacterial species[51].

Based on our sequence data, we have shown insights into the natural genomic evolution of LLOV. Comparing the LLOV genomic sequence data from this study with the reference genome sequence from Spain, the level of LLOV sequence variation is consistent with other filovirus genera. For ebolaviruses and marburgviruses, the GP gene is a hotspot of nucleotide sequence variation[52,53]. The consistency of elevated GP sequence variation compared to other genomic regions in all three genera (*Ebola virus*, *Marburgvirus*, and *Cuevavirus*) suggests similar genetic evolutionary driving forces on this region (e.g., immune evasion). Based on the presented data and considering the previous reports from Spain[14,16], there are multiple reports of LLOV infection in Schreiber's bats, consistent with the idea that this species may serve as a natural reservoir of the virus, however at this point we can only conclude the role of these animals as hosts for the virus in Europe. Despite many remaining questions, we provide evidence and further support for the role of bats in the natural circulation of filoviruses in general. The transmission cycle of LLOV may be complex and potentially different from filoviruses whose enzootic cycle occurs in tropical climates. As a major difference, we present data showing LLOV RNA-positive highly-specialized bat-associated ectoparasites, highlighting the importance of future research regarding the role of these insects in LLOV and other filovirus transmissions. Similar surveillance studies may greatly facilitate the discovery and understanding of novel filoviruses. Most importantly, we have isolated the infectious virus which represents the only filovirus that does not belong to the ebola- or marburg-virus genera, and it is the third successful isolation of a filovirus directly from bats, following MARV and RAVV virus[6], opening the door to further virological studies. Importantly, we show that multiple human cell lines are permissive to LLOV infection, which raises concerns about potential bat to human spillover events with LLOV in Europe and urges immediate pathogenicity and antiviral studies.

## Methods

**Ethics and biosafety statement**. Bat sampling activities on a country-wide scale were approved by the Hungarian Government Office of Pest County under the registration number of PE-KTFO/4384-24/2018. Animal handling was performed by licensed chiropterologists, no animals were harmed during the study, and all ethical standards were followed during the work. All in vitro virus isolation procedures were performed under Biosafety Level 4 conditions within the laboratory of Szentágothai Research Center, University of Pécs, Hungary.

**Animal sampling**. We conducted regular bat colony monitoring during the hibernation period (between November and March) (passive surveillance) at the site of LLOV re-emergence in Hungary (Northeast Hungary, Zemplén Mountains) from 2016 to 2020, collecting only deceased bats, if found. Whole carcasses were frozen and transferred in a dry shipper to the biosafety level 4 (BSL-4) laboratory at the Szentágothai Research Center, Pécs, Hungary, where they were dissected. Lung, spleen, kidney, liver, and brain tissues as well as blood samples were stored at −80 °C until further analyses.

In addition, seven live animal sampling events (active surveillance) were performed between 2018 and 2020, when samples (blood, urine, feces, ectoparasites) were collected from Schreiber's bats (*Miniopterus schreibersii*) and Greater mouse-eared bats (*Myotis myotis*). Bat species identification was performed by trained chiropterologists according to morphological identification keys[54]. Considering conservational aspects, all sampling activities were conducted after the reproduction period of the colony (between August to November). During the live sampling events, serum samples were taken from captured bats after which they were left hanging separately in disposable paper bags (air permeable) for ~2 h, while on-site RT-PCR analysis for the presence of LLOV RNA in from blood samples was performed (as detailed in the *Virus detection* section). This methodology permitted the observation and re-sampling of infected bats. Altogether, 376 bat individuals were sampled between 2016 and 2020 (Supplementary Table). Whole blood (maximum of 50 μL) was taken from the uropatagium vein from each animal by Minivette® POCT (Sarstedt, Germany) disposable microtubes. At the first live animal sampling event (18.09.2018), the blood samples were collected in 1.5-ml Eppendorf tubes, where serum was separated for neutralization assays by low-speed centrifugation (1000 × g) for 5 min. Cell pellets were used for nucleic acid extraction and RT-PCR detection of LLOV RNA. Due to the multipurpose nature of the investigation and the strong limitation of the blood amount, in case of samples less than 8 μL, only LLOV RNA detection was conducted. When more blood was collected (~8–13 μL), only serology was performed. For samples with volumes above 13 μL, both LLOV RT-PCR and serology testing were performed.

Following the first live sampling, we changed the blood sampling methodology in order to provide the possibility of in vitro isolation efforts. During the following six live sampling events (02.08.2019–18.09.2020), freshly obtained blood samples were transferred directly to 200 μL Virus Transport Medium (VTM) (UniTranz-RT 1 mL Transport System, Puritan, USA), and mixed gently by pipetting. A 100 μL portion of each sample was deposited for nucleic acid extraction while the remainder was stored and served as inoculum for virus isolation experiments. When possible, urine, feces, and ectoparasites (ticks, bat flies) were also collected with forceps during bat handling. Feces and urine samples and ectoparasite specimens were immediately individually frozen in liquid nitrogen, and stored at −80 °C until further laboratory processes. Ectoparasites of the *Nycteribiidae* family were identified based on morphological characteristics, identifying parasite species or genus according to generally approved reference keys using a stereomicroscope[55,56]. Tick samples were barcoded by sequencing the mitochondrial 16S rDNA sequence[57].

**Virus detection**. For LLOV RNA detection, we performed RT-PCR tests from all sample types (lung, spleen, kidney, liver, brain tissues and blood from carcasses; blood, urine, feces samples and ectoparasite specimens from live bats). For blood samples in VTM, we carried out on-site LLOV RNA detection in the field, while other samples were transported to the laboratory for testing. These blood samples were further screened with a pan-filovirus nested-RT-PCR, previously used to discover several novel filoviral sequences in China[58].

Tissues, feces, and ectoparasites were homogenized in 100 μL 1x Phosphate Buffered Saline (PBS) using Minilys Personal Homogenizer (Bertin, Germany) with glass beads. The urine samples were complemented with 1× PBS to a total volume of 100 μL. Then, the samples were centrifuged at 16,000 × g for 5 min and 100 μL supernatant was transferred for RNA extraction. The 100 μL blood samples in VTM were directly used for RNA extraction.

RNA was extracted using Direct-Zol RNA MiniPrep (Zymo Research, USA). RNA was then used for RT-PCR using the QIAGEN OneStep RT-PCR kit (Qiagen, Germany) at 50 °C for 30 min, and 95 °C for 15 min, followed by 45 cycles of 95 °C for 15 s, 60 °C for 20 s 72 °C for 40 s (the fluorescence signal was detected during the annealing step). We used the primers FiloAneo: 5′-ARG CMT TYC CAN GYA AYA TGA TGG T-3′ and FiloBNeo: 5′-RTG WGG NGG RYT RTA AWA RTC ACT NAC ATG-3′ and the probe Lloviu-S: FAM-5′-CCT AGA TTG CCC TGT TCA TGA TGC CA-BHQ1-3′[16]. It is noteworthy that a previously described detection method published by our laboratory[15] resulted in a significant number of false positives, which in certain cases we subsequently failed to validate with a pan-filovirus nested-PCR method and sequencing[7]. All experiments were run on the

MyGo Pro PCR system platform (IT-IS Life Science, Ireland) and analysed on the MyGo PCR software (v.3.5.21).

**Virus isolation.** Cell lines used in this study: Cercopithecus aethiops (African green monkey kidney) cells (Vero E6; ATCC® CRL-1586™), Miniopterus sp. kidney cells (SuBK12-08; kindly provided by Ayato Takada, Hokkaido University)[18,35], human neuroblastoma cells (SH-SY5Y; ATCC® CRL-2266™), human hepatocellular carcinoma (Hep G2; ATCC HB-8065™), human non-small cell lung carcinoma (HCC-78; DSMZ ACC 563), human colon carcinoma (HCT 116; DSMZ ACC 581), Tadarida brasiliensis lung cells (Tb 1 Lu; ATCC® CCL-88™).

Vero E6 cells were maintained in Dulbecco's modified Eagle medium (DMEM, Lonza Cat. No: 12-604 F.), SH-SY5Y and Hep G2 cells were maintained in DMEM with 1% of MEM Non-essential Amino Acid Solution 100× (NEAA) (Sigma Cat. No: M7145), SuBK12-08 and Tb 1 Lu in Eagle's Minimum Essential Medium (EMEM, Lonza Cat. No: 12-662 F) with 1% of L-Glutamine (200 mM) (Lonza Cat. No: BE17-605E), HCC-78 in RPMI-1640 (Sigma Cat. No: R5886), HCT 116 in McCoy's 5 A (Lonza, Cat. No: BE12-688F). All culture medium was supplemented with 10% heat inactivated Fetal Bovine Serum (FBS) (Gibco Cat. No: 16140071), and 1% Penicillin/Streptomycin (10,000 U/mL). All cell lines were grown at 37 °C, 5% $CO_2$.

Freshly obtained blood samples from field PCR-tested LLOV RNA-positive bats were frozen in liquid nitrogen and transferred to the BSL-4 laboratory at the Szentágothai Research Center to perform virus isolation experiments. Of note, due to the low body weight of the animals, the blood sample volumes are extremely low, usually below 50 µL. The blood samples (5–50 µL, depending on the original amount) were complemented with cell culture media to a volume of up to 50 µL and used to inoculate cell monolayers seeded in 24-well plates with ~80% confluency. After an 1 h incubation at 37 °C and 5% $CO_2$, the cells were washed once with cell culture media. The cells were monitored daily for cytopathic effects. After 7 days, we performed two freeze-thaw cycles before culture supernatants were transferred to fresh cells. After three blind passages, cells were lysed by the freeze-thaw method, subjected to nucleic acid extraction by Direct-Zol RNA MiniPrep (Zymo Research, USA) and analyzed for the presence of LLOV RNA by RT-PCR using LLOV-specific primers. Viral titers were determined by tissue culture infectious dose ($TCID_{50}$) assays on SuBK12-08 cells using the Spearman-Karber algorithm.

**RNA fluorescence in situ hybridization analysis.** For RNA fluorescent in situ hybridization (FISH) analysis, Vero E6 cells seeded in 8-well chamber slides were mock-infected or infected with LLOV at an MOI of 3. Cells were fixed 1 day post-infection in 10% formalin for at least 6 h. RNA FISH was performed using the RNAscope Multiplex Fluorescent V2 kit (Advanced Cell Diagnostics). Viral RNA was detected using custom-designed probes targeting the negative-sense genomic sequence of the LLOV NP gene (Advanced Cell Diagnostics) and stained with Opal 690 fluorophore (Perkin-Elmer). Staining was performed according to the manufacturer's protocol for adherent cell samples, with the exception of an additional HRP blocking step following signal development of the probes detecting viral mRNA as per the manufacturer's recommendation. Nuclei were stained with kit-supplied DAPI following the manufacturer's protocol. Coverslips were mounted on slides using FluorSave mounting medium, and slides were subsequently stored at 4 °C prior to imaging. Images were acquired at ×60 magnification using a Nikon Ti2 Eclipse microscope and Photometrics Prime BSI camera with NIS Elements AR software.

**Generation of LLOV pseudotyped lentiviruses and neutralization tests.** The LLOV GP gene (GenBank accession JF828358) was inserted into the pCAGGS expression plasmid (kind gift of Prof Ayato Takada). Generation of LLOV and EBOV GP pseudotyped lentiviruses was based on a protocol described previously[59], with co-transfection of three plasmids, pCAGGS-LLOV GP or pCAGGS-EBOV GP, p8.91 (HIV-1 gag-pol) and pCSFLW (luciferase reporter), into HEK293/17 cells, using 1 mg/mL branched Polyethyleneimine (PEI) (Sigma-Aldrich, USA) at a ratio of 1:10 (µg DNA: µL PEI). Pseudotyped lentivirus (PV) supernatant was harvested 48 h later and stored at −80 °C. Titration of PVs was carried out as described elsewhere[60] by twofold serial dilution across white Nunc flat-bottomed 96-well microplates (Thermo Fisher Scientific, USA) in HEK293T/17 target cells ($2 × 10^4$/well), incubated at 37 °C, 5% $CO_2$. After 48 h, the media was removed and discarded; Bright-Glo reagent was added to the plate and incubated at room temperature for 5 min before measuring luminescence on a GloMax 96 luminometer (Promega, USA), with titers given in Relative Light Units (RLU) per mL. For ease of inter-lab comparison, $TCID_{50}$ titers were also obtained using the same methodology but using a fivefold dilution series.

Luminescence values were used to calculate the PV titre ($TCID_{50}$/mL) using the Reed-Muench method[61]. The cumulative number of positive and negative wells for PV infection at each dilution was determined and the percentage calculated for each. The threshold value for a positive well was set at 2.5× the average luminescence value of the cell-only negative controls (Supplementary Fig. 2).

Pseudotyped virus neutralization tests (PVNTs) were performed in white microplates by serial dilution of bat sera (1:40 to 5120; 1:100 to 12,800 or 1:200 to 25,600) with PVs (~$1 × 10^5$ RLU/well or ~100 $TCID_{50}$/well, calculated according to

the titration result) incubated at 37 °C for 1 h. Target HEK293/17 cells were then added and plates read was performed[60,62]. In total, 78 serum samples, mainly from live bats, were tested. Control serum standards (WHO NIBSC15/262 & NIBSC15/282 antibody standards) obtained from EBOV disease convalescent patients were used, as no LLOV-positive serum samples were available. Data was normalized to % reduction in luminescence with respect to the average RLU of cell only (100% neutralization) and PV only (0% neutralization) controls and fitted into a non-linear regression model (log [inhibitor] vs. normalized response, variable slope). $IC_{50}$ antibody titres were calculated using Prism 8 software. Average values of two independent experiments are indicated unless otherwise stated. The cut-off for PVNT positivity was determined using $IC_{50}$ values from samples where the total antibody-mediated reduction in RLU was <20% of the PV + no serum control (i.e., 0% neutralization) or when a neutralization of <70% was achieved. The mean of these values was calculated and a cut-off was defined for values higher than the mean + three times the standard deviations[63–65] (Supplementary Fig. 5).

**Amplicon-based nanopore sequencing.** We developed an amplicon sequencing method based on previous protocols[66,67] that is able to amplify the LLOV-specific gene regions in two parallel multiplex PCR reactions[37]. LLOV RNA-positive samples LLOV_378 and LLOV_378P1 were used for sequencing. cDNA preparation from RNA samples was conducted with SuperScript IV (Invitrogen, USA) with random hexamers. Amplicons were generated from cDNA with Q5 Hot Start HF Polymerase (New England Biolabs, USA), with three different primer sets in parallel pools (namely: LLOV_400bp pool 1 and 2, LLOV_500bp pool 1 and 2, LLOV_2000bp pool 1 and 2) (Supplementary Fig. 2). 2.5 µL amplicons from each pool belonging to the same primer set (Primer set: 400 or 500 or 2000) were diluted in 45 µL nuclease-free water before the end repair and dA tailing were performed with the NEBNext Ultra II End Repair/dA-Tailing Module (New England Biolabs, USA). 1.5 µL end-prepped DNA was transferred to the next reaction directly and the barcodes derived from EXP-NBD196 (Oxford Nanopore Technologies, UK) were ligated with NEBNext Ultra II Ligation Module (New England Biolabs, USA). After the clean up of the pooled different barcoded samples jointly with Ampure XP beads (Beckman Coulter, USA), the AMII sequencing adapters were ligated with NEBNext Quick Ligation Module. The final library was quantified with Qubit dsDNA HS Assay Kit (Invitrogen, USA) on a Qubit 3 fluorometer. 25 ng final libraries were loaded onto a R9.4.1 (FLO-MIN106D) flow cell and were sequenced for 48 h[37].

**Bioinformatic pipeline.** The raw sequencing data were basecalled using guppy (ONT guppy v4.4.2.) high accuracy basecaller algorithm (dna_r9.4.1_450bps_hac config file). Demultiplexing and trimming of barcodes were performed also with guppy using default parameters of 'guppy_barcoder' runcode. The ONT guppy software was used under Ubuntu Linux 18.04. Because ARTIC-like protocols generate chimeric reads, we performed length filtering with appropriate lengths specific to the used primer set (LLOV_400bp: 350-600 bp, LLOV_500bp: 440–700 bp, LLOV_2000bp: 1400–2700 bp). Primers were trimmed with the BBDuk (v38.84) Geneious Prime (v2021.1.1) plugin. To generate a consensus sequence, the processed reads from all samples were mapped to the LLOV reference genome (NCBI accession number: NC_016144) with the usage of MiniMap 2.17[68]. The generated consensus sequences were manually checked for basecalling errors.

**Sequencing of the virus isolate.** Prior to Nanopore sequencing, virus isolates were exposed to a generally used enrichment protocol[69,70]. Nucleic acid was extracted from samples using the Direct-zol RNA Miniprep Plus Kit (Zymo Research, USA). Samples were then subjected to the Sequence Independent Single Primer Amplification (SISPA) approach with minor modifications to the previously described protocol[70]. cDNA amplification was performed by SuperScript IV Reverse Transcriptase (Thermo Fisher scientific) with dNTPs (10 µM) and 2 µM K-8N primer following the manufacturer's instructions. In order to convert double stranded cDNA from first strand cDNA, samples were subjected to Klenow reaction. Thereafter, the products underwent a purification step using Agencourt AMPure XP beads (Beckman Coulter). Following that, ds cDNA was amplified by Q5® High-Fidelity DNA Polymerase (NEB). Amplified cDNA was purified by Agencourt AMPure XP beads (Beckman Coulter) and quantified using a Qubit dsDNA BR Assay kit (Thermo Fisher Scientific). The double stranded fragments were end prepped with NEBNext Ultra II End Repair/dA-Tailing Module (New England Biolabs, USA) and were barcoded with EXP-NBD196 (Oxford Nanopore Technologies, UK) kit. The final library was sequenced on a R9.4.1 (FLO-MIN106D) flow cell.

The raw sequencing data were basecalled using guppy (ONT guppy v5.0.7.) super-accuracy basecaller algorithm (dna_r9.4.1_450bps_sup config file). Demultiplexing and trimming of barcodes were performed also with guppy using default parameters of 'guppy_barcoder' runcode. The ONT guppy software was used under Ubuntu Linux 18.04. Porechop v0.2.4 was used to trim primer sequences used for SISPA amplification. To generate a consensus sequence, Medaka v1.4.2 was used to map the trimmed reads against the LLOV reference genome (NCBI accession number: NC_016144) and perform variant calling. Extension of the genome ends was done by manually filling in the observed

overhangs based on a mapping performed using the Long Read Support tool in CLC Genomics workbench v20.0.4. Extension was stopped when read coverage fell below 20×.

**Phylogenetic tree caption**. Prior to the phylogenetic tree implementation, 22 full length sequences belonging to the family *Filoviridae* were aligned in the MAFFT webserver (version 7 https://mafft.cbrc.jp/alignment/server/) using default parameters. Subsequently, a best substitution model selection was performed in IQ-tree webserver (http://iqtree.cibiv.univie.ac.at/) and thus a Bayesian phylogeny tree was generated using BEAST v1.10.4[71] under a GTR + G + I substitution model, assuming a constant population size and a strict molecular clock (uniform rates across branches). The MCMC chains were run for 10,000,000 iterations and sampled each 1,000th generation. The effective sampling size was checked in Tracer software (>200). Thereafter, the Maximum clade credibility tree (MCC) was generated using TreeAnnotator and edited in the iTol webserver (https://itol.embl.de/).

**Statistics and reproducibility**. Depending on the available amount of bat sera different measurement and dilution strategies were applied: measured as duplicate with repeated experiment—bat ♯1, ♯169, ♯170, ♯99, ♯115, ♯130; measured in duplicate with single experiment—bat ♯2, ♯102, ♯138; measured in single in one experiment—bat ♯98, ♯110, ♯118, ♯143. All dilutions, including standard sera were 1/40, 1/80, 1/160, 1/320, 1/640, 1/1280, 1/2560, 1/5120 except for bat ♯143 where 1/100 starting dilution was applied.

In vitro isolation experiments were performed in triplicates with same results. The RNA FISH analysis was performed once. The probes were evaluated with recombinant LLOV and reliably detected positive and negative-sense recombinant LLOV RNA in infected cells[21].

**Reporting summary**. Further information on research design is available in the Nature Research Reporting Summary linked to this article.

## Data availability

The genomic sequence data in this study have been deposited in the NCBI GenBank database under accession codes: MW775010, MW775011 and MZ541881. Tick samples: OL795929-OL795963 [https://www.ncbi.nlm.nih.gov/nuccore/?term=ixodes+lloviu +hungary]. The background data of sampled animals, such as sex and collection dates are provided in the Supplementary Data 1. Sequencing protocol and materials are listed here: https://www.protocols.io/view/lloviu-cuevavirus-sequencing-protocol-bmz3k78n.html. Source data are provided with this paper.

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

## Acknowledgements
The project was supported by the National Research, Development and Innovation Office (grant numbers: NKFIH FK131465 (G.K.), KH129599 (G.K., F.J.). G.K. was supported by the János Bolyai Research Scholarship of the Hungarian Academy of Sciences. G.E.T. and Z.L. were supported by the Biological and Sportbiological Doctoral School of the University of Pécs, Hungary. The SuBK12-08 cell line was obtained from the scientists of Hokkaido University Center for Zoonosis Control and the University of Zambia. The authors wish to thank professor Ayato Takada for sharing the SuBK12-08 cells. This work was supported by NIH grants R21 AI137793 (E.M.) and R01 AI133486 (E.M.). E.L.S. was supported by NIH training grant T32 HL007035.

## Author contributions
G.K., G.E.T., M.M.N., S.S., B.Z., Z.L., Á.N, C.I.P., G.D., F.F. conceived the laboratory work. G.K., G.E.T., S.A.B., T.G., P.E., Z.L., Á.N., C.I.P., G.D., K.K. conducted field work. M.M.N., S.S., N.T., E.W. designed and conducted the neutralization experiments. S.A.B., T.G., P.E. handling of bats during field work. P.E., T.S. ectoparasite identification. G.K., M.M., F.F. conceived the in vitro isolation experiments. E.L.S. performed the RNA FISH experiments. G.E.T., B.A.S., A.J.H., S.Z., G.K., P.M, B.V. sequencing and genomic analysis. B.A.S., M.M.N. visualization of results. G.K., F.J., S.S. conceptualization and supervision. G.K. wrote the paper. G.E.T., M.M.N., S.S., N.T., E.W., S.A.B., T.G., Á.N., F.J., A.J.H., E.M. edited the paper with contributions from all other authors. G.K., S.S., E.M., Á.N., F.J. provided supervision. All authors approved the paper.

## Funding

## Competing interests
The authors declare no competing interests.
