## [Peer Review File · Nature Communications]

Isolation of infectious Lloviu virus from Schreiber's bats in HungaryReviewers' Comments:

Reviewer #1:

Remarks to the Author:

Kemenesi et al. report the detection and isolation of LLOV from Schreiber's bats in Europe. The study is important from a disease ecology and virology perspective and reports the isolation of LLOV from bats for the first time. The study also demonstrates that wildtype LLOV is capable of infecting other mammalian cells identifying the potential zoonotic potential of this virus. These are important observations that will direct future research on emerging filoviruses. However, the neutralization assay needs further validation to rule out potential cross-reactivity with other known and unknown filoviruses in Schreiber's bats. Furthermore, the lack of long-term follow-up studies in tagged bats makes it difficult to conclude if these bats are indeed asymptomatic reservoirs of LLOV. More detailed comments are below.

1. Although the authors acknowledge that limited sera were available from the bats, the current neutralization data does not rule out the presence of cross-reactive antibodies that may neutralize LLOV. It is critical to demonstrate that the sera do not react with GPs from other filoviruses that these bats may be exposed to. This is my greatest concern about this manuscript as a large proportion of information is derived from this assay with the presumption that the assay does not detect other cross-reactive filovirus antibodies.

2. Was there a difference in the amount of virus detected in dead vs. live bats? LLOV infection was previously casually associated with lethal infection in bats. Thus, it is intriguing to know if the extent of virus infection and virus spread is responsible for lethal outcomes in some bats. Perhaps better highlight the qRT-PCR results in your Discussion? This is an important observation.

3. It is difficult to conclude from the data that infected but asymptomatic bats would have remained asymptomatic during infection. The infected bats were not followed up till the resolution of infection. With the current data, it is difficult to rule out the possibility that the infected bat would eventually succumb to infection. To demonstrate that this bat species is a reservoir of LLOV, it is important to demonstrate the 'lack of' Koch's postulates or demonstrate in a controlled setting that infected bats can shed the virus, followed by resolution of viremia without demonstrating significant morbidity or mortality. Now that the virus has been isolated, it might be possible to demonstrate infectivity and pathogenesis in wild-caught or captive bats (if available).

4. Overall, the data are intriguing and the report of the first isolation of LLOV is certainly important, but I would advise against drawing far-fetched conclusions about the reservoir status of these bat species. Controlled studies would be required to identify the pathogenesis of LLOV in these bats, along with determining the possible role of these bat species as reservoirs of LLOV. The authors acknowledge the limitations of their study in the Discussion. This should perhaps be highlighted across the text.

5. Dead bats with LLOV RNA were identified during hibernation. Is it possible that hibernation facilitates virus replication and thus, pathogenesis? What about hibernation and impact on antiviral immunity in these bats? These might be worth discussing or mentioning.

6. The detection of LLOV RNA in bat ectoparasites in this study does not prove that the ectoparasites also transmit the virus. The authors have acknowledged this in their Discussion. However, developing a trial to demonstrate transmission or lack of transmission of LLOV via hematophagous vectors would add more significance to the findings. If the bat ectoparasites are dead-end hosts, this would present a bottleneck for virus spillover, which would significantly lower the risk of LLOV spill over via vectors. The lack of viral RNA in urine and feces of these bats further suggest that perhaps LLOV is not spread via the fecal-oral route (or perhaps the sampling time/strategy failed to capture virus shedding). However, as mentioned above, controlled challenge studies would be required to draw these

conclusions.

Reviewer #2:

Remarks to the Author:

The manuscript by Kemenesi et al provides only the second description of the isolation of a filovirus from any bat species. Great to see this type of work being carried out and great to see a study that correlates ectoparasites with positive bats. Just a few minor comments.

1. p 5, line 123-132: the last two paragraphs of the introduction are unnecessary and seem out of place. The ectoparasite finding can be added to the earlier paragraph.

p5, Results: please provide some description/observations on the bats. Were they apparently healthy? Mixed sex? All adults or juveniles? Any other observations? Why were there bat carcasses and what condition were they in? There seems to be some information later on this so move it to right at the beginning of the results.

p6. line 149: How many carcasses were tested?

p7: were the RT-PCR products sequenced?

p9, line 213: This appears to be the first instance that the SuBK12-08 cells are mentioned. Please provide more detail on these cells (ie. species, tissue derived from, primary/immortalised).

p9, line 213: was LLOV_378 the only blood sample passaged on the cells?

p9. was there any observable CPE when the virus was passed on the different cell lines.

p12, line 272: The authors describe detection of the virus in asymptomatic bats, yet there were carcasses that were seropositive and positive by RT-PCR. Any evidence of how these bats died? There are contradictory statements in this paragraph - describing evidence of asymptomatic animals as evidence of their reservoir status and then saying some bats may have died of the infection.

Was virus isolation attempted on any of the ectoparasites?

Reviewer #3:

Remarks to the Author:

In this manuscript by Kemenesi, the authors report the results of a comprehensive 3-4 year effort (2016-19) to survey a colony of Schreiber's bats (*Miniopterus schreibersii*) in Hungary for the presence of Lloviu virus (LLOV), a filovirus first described almost 20 years ago in Spain but never isolated. Here, the authors report the first isolation of this virus directly from bat blood. This is a major achievement, not only for being the first isolation of a European filovirus but the first isolation of a filovirus not in the Ebola- or Marburgvirus genera. The authors use a pseudotype virus neutralization assay to show that seroprevalence in the colony gradually increases during the year, peaking in late summer/fall, coincident with the influx of weaned and independent juveniles into the population. They also show that some hematophagous ectoparasites tested weakly positive, ~100x (based on Ct values) lower than the level in the blood LLOV-positive ectoparasites were only found on positive bats. The finding of LLOV positive ectoparasites raises the possibility of their involvement in mechanical virus transmission and contrasts the lack of such detections in parasites associated with Egyptian rousette bats, a known reservoir for marburgviruses. Finally, the authors show that various human cell

lines are permissive to LLOV, raising the possibility that humans could be infected, though no human infections have been reported. Overall, this is a very well written report with convincing data that would be of interest to a broad audience and even the public at large.

Minor queries are listed below:

1. Lines 69, 121, Two marburgviruses, MARV and RAVV (Ravn virus) were isolated from Egyptian rousette bats, therefore making LLOV isolation the 3rd filovirus isolated from bats.
2. Can the authors comment on if they've tried to correlate the pseudovirus neutralization test and a live virus neutralization test now that they have an LLOV isolate. This might not be possible with the limited bat sera available, but perhaps with sera raised in another animal inoculated with LLOV.
3. Can the authors comment on how LLOV seropositivity and PCR positivity correlates with bat age (juvenile vs adult), sex, breeding condition etc. This data, if known, could be added to Table 1.
4. The fact that the highest levels of antibody were found in dead bats with 2/4 also being PCR positive in lung/spleen, suggests Ab in these bats might not be very effective? Is it suggestive the bats don't clear the virus? Can the authors comment?
5. Were any of the dead bats tested for co-infection by other agents, bacterial or viral? Any other testing done, histopathology etc on tissues, to determine cause of death?
6. The authors should sequence the rRNA if they cannot identify the tick to the species level by morphology. Please see the following publication that might help:
<https://www.liebertpub.com/doi/10.1089/vbz.2006.6.152>
7. Oral swabs were notably absent from the sampling regimen. Any particular reason? MARV, RAVV and BOMV have all been detected in oral secretions.
8. Can the authors comment on the rough size of the population at the study site and the degree to which it fluctuates over the year? Is there a resident population during winter, how big? Population in summer, how big or fold increase over winter size? Is it known roughly what percent of adult females get pregnant each year?

Response to the reviewers' comments

Following the recommendation of reviewers and the editor, we performed additional experiments to validate our results.

- *Neutralization assay validation (Rev 1):*

The cross-reactivity of different filovirus anti-GP nAbs were previously reported in other studies. However the nature of cross-reactivity is complex and based on relevant literature it is the most closely related filovirus species that are most likely to cross react with each other (see refs below):

Bentley, E.M.; Richardson, S.; Derveni, M.; Rijal, P.; Townsend, A.R.; Heeney, J.L.; Mattiuzzo, G.; Wright, E. Cross-Neutralisation of Novel Bombali Virus by Ebola Virus Antibodies and Convalescent Plasma Using an Optimised Pseudotype-Based Neutralisation Assay. *Trop. Med. Infect. Dis.* **2021**, *6*, 155. <https://doi.org/10.3390/tropicalmed6030155> and

Hargreaves, A.; Brady, C.; Mellors, J.; Tipton, T.; Carroll, M.W.; Longet, S. Filovirus Neutralising Antibodies: Mechanisms of Action and Therapeutic Application. *Pathogens* **2021**, *10*, 1201. <https://doi.org/10.3390/pathogens10091201>

In addition to this, GP was found to be the most specific antigen, whilst NP and VP40 antigens were found to be more cross-reactive. In a second study, authors also found the GP antibody response as the most specific:

Natesan M, Jensen SM, Keasey SL, Kamata T, Kuehne AI, Stonier SW, Lutwama JJ, Lobel L, Dye JM, Ulrich RG. Human Survivors of Disease Outbreaks Caused by Ebola or Marburg Virus Exhibit Cross-Reactive and Long-Lived Antibody Responses. *Clin Vaccine Immunol.* 2016 Aug 5;23(8):717-24. doi: 10.1128/CI.00107-16

Kamata T, Natesan M, Warfield K, Aman MJ, Ulrich RG. Determination of specific antibody responses to the six species of ebola and Marburg viruses by multiplexed protein microarrays. *Clin Vaccine Immunol.* 2014;21(12):1605-1612. doi:10.1128/CI.00484-14

Bat-related data is deficient, although in a study, very little cross-reactivity was seen between filovirus genera using bat sera and an IgG indirect ELISA methodology:

Schuh, AJM, Amman BT, Sealey TS, Flietstra TD, Guito JC, Nichol ST, Towner JS. Comparative analysis of serologic cross reactivity using convalescent sera from filovirus experimentally vaccinated fruit bats. *Scientific Reports* **2021**, *9*, 6707. <https://doi.org/10.1038/s41598-019-43156-z>

In terms of phylogenetic position and functionality, the currently known most closely related relative of LLOV is Ebola virus. Therefore we used the last remnants of LLOV nAb positive bat sera to test the cross-reactivity in an EBOV Gp pseudotyped neutralization assay. We detected no cross-reactivity in the experiment, which accords with the finding that EBOV convalescent sera did not react to LLOV GP pseudotypes in our neutralisation test. We have now included the former results of these experiments as Extended data Figure 4. A and B. Unfortunately at the moment we do not have any additional sera samples to do further testings of cross reactivity, though this will be incorporated in future studies.

We added more discussion about this topic to the Discussion part.

- *Improve the characterization of the isolated virus in vitro (Rev 2):*

To further verify the presence of a replicative LLOV in the isolate, we performed an RNA fluorescence in situ hybridization (RNA-FISH) analysis using probes targeting the negative-sense genomic sequence of the LLOV NP gene. We verified the presence of replicative LLOV and added a novel figure to the manuscript presenting the cytopathogenic effect of the virus on SuBK12-08 cell line along with the RNA FISH results on Vero E6 cell line. (Fig. 3.)

We have also attached two additional visual elements (movies) about the progress of cytopathogenic effect on Vero E6 and SuBK12-08 cell monolayers (Supplementary Movies).

- *Provide additional information/data on bats and ectoparasites (Rev 1, Rev 3):*

We performed 'barcoding reactions' on the LLOV positive tick samples, sequencing the mitochondrial 16S rDNA gene fragment. 33 samples were identified as *Ixodes simplex*, whilst the remaining two were identified as *Ixodes vespertilionis* using the methods of Lv et al., 2014. We added this information into the manuscript Supplementary table, along with the related accession numbers: OL795929-OL795963

All the bat-related data are also summarized within the Supplementary Data Table.

Reference:

Lv J, Wu S, Zhang Y, et al. Assessment of four DNA fragments (COI, 16S rDNA, ITS2, 12S rDNA) for species identification of the Ixodida (Acari: Ixodida). *Parasit Vectors*. 2014;7:93. Published 2014 Mar 3. doi:10.1186/1756-3305-7-93

- *Asymptomatic infection (Rev 1):*

We have modified the manuscript in accordance to the reviewers opinion and the editorial request. We fully agree with the remarks, at this point we cannot draw strong conclusions about the disease progression in these bats during LLOV infection. We plan to integrate unique identification for the individual bats in our future studies. Nevertheless simply based on the currently available data we moderated the statement on asymptomatic infection.

Detailed response to each point:

Reviewer 1

Kemenesi et al. report the detection and isolation of LLOV from Schreiber's bats in Europe. The study is important from a disease ecology and virology perspective and reports the isolation of LLOV from bats for the first time. The study also demonstrates that wildtype LLOV is capable of infecting other mammalian cells identifying the potential zoonotic potential of this virus. These are important observations that will direct future research on emerging filoviruses. However, the neutralization assay needs further validation to rule out potential cross-reactivity with other known and unknown filoviruses in Schreiber's bats. Furthermore, the lack of long-term follow-up studies in tagged bats makes it difficult to conclude if these bats are indeed asymptomatic reservoirs of LLOV. More detailed comments are below.

Response: Thank you for your valuable work during the revision process. We agree with the necessity of further validation experiments for the neutralization work. In accordance with this request we performed additional experiments with the last remnants of the LLOV nAb positive serum samples. We did not detect the cross reactivity of EBOV sera (WHO standard) on LLOV PVNA or the cross-reactivity of LLOV positive sera in EBOV PVNA. Unfortunately all the remaining sera were used for these efforts. Please also read the first point of this response letter for more details.

We plan to extend our research portfolio with unique tagging of bats in the future. We already started the permission process for these activities during the upcoming years.

1. Although the authors acknowledge that limited sera were available from the bats, the current neutralization data does not rule out the presence of cross-reactive antibodies that may neutralize LLOV. It is critical to demonstrate that the sera do not react with GPs from other filoviruses that these bats may be exposed to. This is my greatest concern about this manuscript as a large proportion of information is derived from this assay with the presumption that the assay does not detect other cross-reactive filovirus antibodies.

Response: We understand the concern about cross-reactivity. However there are extremely limited possibilities, considering that the availability of nAb positive sera from other filovirus infections is limited to a small group within the filoviridae family. We performed additional cross-reactivity tests as detailed above.

In addition to this, we do not expect the presence of other filoviruses other than LLOV in European bats. This is based on the numerous European studies, involving viral metagenomics or targeted surveillance where there

were no any indication for the presence of other filoviruses than LLOV in Europe so far. Please find a recent literature review about the European bat virological examinations, as a support to this statement: <https://www.mdpi.com/2076-393X/9/7/690>

During our LLOV-related research, we performed targeted, pan-filovirus screening on the samples of this study in order to verify the specificity of real-time RT-PCR results (Supplementary Data Table). We used the pan-filoviral nested-PCR screening system, originally published by He et al. 2015. With this method we were able to verify the real-time RT-PCR positives, and apart from those samples we found no other positives. Based on these data we can conclude that there are no evidence for the circulation of other filoviruses in this colony so far.

We added more discussion about this topic to the Discussion part.

References:

He B, Feng Y, Zhang H, et al. Filovirus RNA in Fruit Bats, China. Emerg Infect Dis. 2015;21(9):1675-1677. doi:10.3201/eid2109.150260

2. Was there a difference in the amount of virus detected in dead vs. live bats? LLOV infection was previously casually associated with lethal infection in bats. Thus, it is intriguing to know if the extent of virus infection and virus spread is responsible for lethal outcomes in some bats. Perhaps better highlight the qRT-PCR results in your Discussion? This is an important observation.

Response: We observed 1-3 order of magnitude difference in the genomic copies / mL between the live and dead animal samples. The viral load in live bats was higher than the observation in dead animals. Although we believe it is not informative to draw conclusions from these results: 1. sample numbers are low, we analysed three dead bat samples and four live bat samples 2. the starting amount for the nucleic acid extraction may differ significantly, it is hard to assess the tissue amount from the dead bats (obviously more than the blood but still hard to exactly assess) and compare it with the amount of blood. 3. Also the condition of bat carcasses may effect the detected viral load and we have no information about the incubation period under cave conditions.

If we consider that the starting material was higher from dead bats and we still observed lower copy numbers compared to the live bat-blood samples, we can still conclude a lower viral load in the tissue of dead animals than the blood of live animals. It raises another concern – since there is no extended data on tissue tropism of LLOV, there are more data needed in terms of pathogenicity and infection progress to assess these genomic copy number-related results.

We added slight modification, pointing to the current observations.

3. It is difficult to conclude from the data that infected but asymptomatic bats would have remained asymptomatic during infection. The infected bats were not followed up till the resolution of infection. With the current data, it is difficult to rule out the possibility that the infected bat would eventually succumb to infection. To demonstrate that this bat species is a reservoir of LLOV, it is important to demonstrate the 'lack of' Koch's postulates or demonstrate in a controlled setting that infected bats can shed the virus, followed by resolution of viremia without demonstrating significant morbidity or mortality. Now that the virus has been isolated, it might be possible to demonstrate infectivity and pathogenesis in wild-caught or captive bats (if available).

Response: We fully agree with this remark, and already made modifications in the manuscript to tone down the asymptomatic infection part. We are grateful for the suggestion of the reviewer and we already started the permission process to use unique identifier tags on these bats in the future – although this process takes time and also the results of capture-mark-recapture surveillance can be presented in a future work.

We understand the strong need for unequivocal proof for the reservoir role of this species, but there are several limiting circumstances to conduct virus challenge experiments on these bats. All bat species in Europe are strictly protected under the Flora, Fauna, Habitat Guidelines of the European Union (92/43/EEC) and the Agreement on the Conservation of Populations of European Bats (www.eurobats.org). This particular species is also strongly affected by the disturbance of colony sites. We believe that we used all possibilities to give the most clear picture about the possible reservoir role and raised several more questions, which may be solved in future studies.

4. Overall, the data are intriguing and the report of the first isolation of LLOV is certainly important, but I would

advise against drawing far-fetched conclusions about the reservoir status of these bat species. Controlled studies would be required to identify the pathogenesis of LLOV in these bats, along with determining the possible role of these bat species as reservoirs of LLOV. The authors acknowledge the limitations of their study in the Discussion. This should perhaps be highlighted across the text.

Response: Thank you for this suggestion, we made modifications accordingly, in multiple places throughout the text we toned down the reservoir role conclusions and indicated these bats as currently known hosts for LLOV. We mention these results as pointing towards the possible reservoir role of these bats and highlight the necessity of future studies to verify this.

5. Dead bats with LLOV RNA were identified during hibernation. Is it possible that hibernation facilitates virus replication and thus, pathogenesis? What about hibernation and impact on antiviral immunity in these bats? These might be worth discussing or mentioning.

Response: Thank you for your valuable comment, we added some discussion about this topic into the manuscript. There are a few previous studies reporting the effects of hibernation in viral infections in bats. Reactivation of latent infection with herpesviruses were reported and also prolonged coronavirus replication for four months in the lungs – pointing to altered immune mechanisms during this period which may ultimately affect viral replication and pathogenesis as well. The knowledge about LLOV infection is strongly limited at this point but involving the hibernation period in future examinations may reveal important results and maybe finally lead to a better understanding on the lethal or non-lethal nature of LLOV infection in bats. Mammalian hibernation is connected with several alternation in homeostasis, including altered immune system attributes. It was observed in multiple mammalian species.

References: Carey, H. V., Andrews, M. T., Martin, S. L. (2003) Mammalian hibernation: cellular and molecular responses to depressed metabolism and low temperature. Physiol. Rev. 83, 1153–1181. Subudhi S, Rapin N, Misra V. Immune System Modulation and Viral Persistence in Bats: Understanding Viral Spillover. Viruses. 2019;11(2):192. Published 2019 Feb 23. doi:10.3390/v11020192

6. The detection of LLOV RNA in bat ectoparasites in this study does not prove that the ectoparasites also transmit the virus. The authors have acknowledged this in their Discussion. However, developing a trial to demonstrate transmission or lack of transmission of LLOV via hematophagous vectors would add more significance to the findings. If the bat ectoparasites are dead-end hosts, this would present a bottleneck for virus spillover, which would significantly lower the risk of LLOV spill over via vectors. The lack of viral RNA in urine and feces of these bats further suggest that perhaps LLOV is not spread via the fecal-oral route (or perhaps the sampling time/strategy failed to capture virus shedding). However, as mentioned above, controlled challenge studies would be required to draw these conclusions.

Response: Thank you for your valuable comment. Controlled infection studies of bat flies is extremely challenging, since they cannot be managed without the host. Regarding the protected nature of bat hosts, we were not able to start related experiments. However we included all possibilities into the discussion part and reflected on these results as needs to be further verified and investigated in the future. During the revision process an excellent paper was published with this type of research from South Africa, where the authors tested the vector potential of bat flies in the transmission of Marburg virus between Rousettus aegyptiacus bats. It is hard to compare those result to our samples and to tempered climate circumstances, although that paper gives a perfect baseline for such studies. We included some more discussion into the manuscript and also included this novel reference. Unfortunately, considering the above detailed nature protection guidelines of bats in the EU we have no chance to perform similar studies. We plan to continue with field observations with stronger focus of the vectors in the future.

Reviewer 2:

The manuscript by Kemenesi et al provides only the second description of the isolation of a filovirus from any bat species. Great to see this type of work being carried out and great to see a study that correlates ectoparasites with positive bats. Just a few minor comments.

1. p 5, line 123-132: the last two paragraphs of the introduction are unnecessary and seem out of place. The ectoparasite finding can be added to the earlier paragraph.

Response: Thank you for your comment and your valuable work during the revision process. We deleted the repetitive paragraph and modified the last one, referring to the sequencing method which we developed and used in this study – we believe that it is a significant part of the manuscript.

p5, Results: please provide some description/observations on the bats. Were they apparently healthy? Mixed sex? All adults or juveniles? Any other observations? Why were there bat carcasses and what condition were they in? There seems to be some information later on this so move it to right at the beginning of the results.

Response: Thank you for the practical suggestion, we made modifications following your guideline.

p6. line 149: How many carcasses were tested?

Response: Thank you for your comment, we clarified the numbers and also made reference for the full table where we present all related data.

p7: were the RT-PCR products sequenced?

Response: real-time RT-PCR products were not sequenced, but we validated all PCR results with a secondary method, published by He et al., 2015. This nested RT-PCR system was demonstrated to be able to detect a variety of viruses within the Filoviridae family. Only the real-time RT-PCR positive samples were positive with this secondary method, confirming our results and also pointing out the absence of other filoviruses other than LLOV within the investigated colony (see comments from Reviewer 1). We indicated the nested-RT-PCR method in the manuscript materials and methods section and also in the Supplementary Data Table.

References:

He B, Feng Y, Zhang H, et al. Filovirus RNA in Fruit Bats, China. Emerg Infect Dis. 2015;21(9):1675-1677. doi:10.3201/eid2109.150260

p9, line 213: This appears to be the first instance that the SuBK12-08 cells are mentioned. Please provide more detail on these cells (ie. species, tissue derived from, primary/immortalised).

Response: We added more information into this paragraph.

p9, line 213: was LLOV_378 the only blood sample passaged on the cells?

Response: Yes, that was the only remaining stock of a positive sample at the time we received the SuBK12-08 cell line. Before that we used all the other samples on the previously mentioned cell lines (Vero E6 and Tb1-Lu) – mostly these activities consumed the amount of available samples. We made minor clarification in the text to make it clear.

p9. was there any observable CPE when the virus was passed on the different cell lines.

Response: Thank you for your question. Yes there were massive CPE observed on SuBK12-08 and Vero E6 cell line as well. We included two additional videos as Supplementary Movies, made with CytoSMART live cell imaging system – on these videos we present the CPE process on SuBK12-08 and Vero E6 cell lines. As we made additional experiments to support the isolation results, we added a novel figure into the manuscript, representing the CPE on SuBK12-08 cell line along with the FISH results on Vero E6 cells.

The other cell types, which we present as a Supplementary Information (SH-SY5Y, HepG2, HCC78, HCT116), were not checked visually, there was no intention for that during the experiment we only relied on viral titer differences. Considering the current role of our laboratory in the national coronavirus research efforts we were not able to repeat those isolation experiments to visually document CPE. We believe that the focus of our study is mostly related to natural transmission patterns and the ecology of the virus, we hope the reviewers can agree with the currently available data which is included and which was extended during the revision process. We have future plans to present detailed CPE and cell-specific viral replication kinetics data. We included the reference of videos into the manuscript text.

p12, line 272: The authors describe detection of the virus in asymptomatic bats, yet there were carcasses that were seropositive and positive by RT-PCR. Any evidence of how these bats died? There are contradictory statements in this paragraph - describing evidence of asymptomatic animals as evidence of their reservoir status and then saying some bats may have died of the infection.

Response: We made several changes and clarifications within the manuscript regarding this topic. We hope it is now more accurate and gives a clearer picture about the current knowns and unknowns along with future research directions.

Was virus isolation attempted on any of the ectoparasites?

Response: Unfortunately we used all ectoparasite materials during the nucleic-acid extraction procedures. Following the current results which are presented in our manuscript, we plan to give a strong focus towards the ectoparasites in future examinations, including in vitro isolation studies.

Reviewer 3:

In this manuscript by Kemenesi, the authors report the results of a comprehensive 3-4 year effort (2016-19) to survey a colony of Schreiber's bats (*Miniopterus schreibersii*) in Hungary for the presence of Lloviu virus (LLOV), a filovirus first described almost 20 years ago in Spain but never isolated. Here, the authors report the first isolation of this virus directly from bat blood. This is a major achievement, not only for being the first isolation of a European filovirus but the first isolation of a filovirus not in the Ebola- or Marburgvirus genera. The authors use a pseudotype virus neutralization assay to show that seroprevalence in the colony gradually increases during the year, peaking in late summer/fall, coincident with the influx of weaned and independent juveniles into the population. They also show that some hematophagous ectoparasites tested weakly positive, ~100x (based on Ct values) lower than the level in the blood LLOV-positive ectoparasites were only found on positive bats. The finding of LLOV positive ectoparasites raises the possibility of their involvement in mechanical virus transmission and contrasts the lack of such detections in parasites associated with Egyptian rousette bats, a known reservoir for marburgviruses. Finally, the authors show that various human cell lines are permissive to LLOV, raising the possibility that humans could be infected, though no human infections have been reported. Overall, this is a very well written report with convincing data that would be of interest to a broad audience and even the public at large.

Response: Thank you for your valuable work during the revision process.

Minor queries are listed below:

1. Lines 69, 121, Two marburgviruses, MARV and RAVV (Ravn virus) were isolated from Egyptian rousette bats, therefore making LLOV isolation the 3rd filovirus isolated from bats.

Response: Thank you for this remark, we corrected accordingly throughout the manuscript.

2. Can the authors comment on if they've tried to correlate the pseudovirus neutralization test and a live virus neutralization test now that they have an LLOV isolate. This might not be possible with the limited bat sera available, but perhaps with sera raised in another animal inoculated with LLOV.

Response: Thank you for the suggestion. We do not operate an animal facility within the BSL-4 laboratory, therefore we do not have the option for animal immunization. In future studies, involving other laboratories and sharing the isolate with them would open the possibility for such experiments.

3. Can the authors comment on how LLOV seropositivity and PCR positivity correlates with bat age (juvenile vs adult), sex, breeding condition etc. This data, if known, could be added to Table 1.

Response: Thank you for your valuable comment. In the case of microbats it is extremely hard to assess the age correctly. We did not collect this data during the fieldwork, considering the questionable reliability.

4. The fact that the highest levels of antibody were found in dead bats with 2/4 also being PCR positive in lung/spleen, suggests Ab in these bats might not be very effective? Is it suggestive the bats don't clear the virus? Can the authors comment?

Response: Following the previous comments from other reviewers we already included some more discussion about the possible effect of hibernation period to viral infection progress. However it is hard to conclude anything based on the currently available data – we need more field observations in the upcoming years, more focused on this period to give prompt answers.

5. Were any of the dead bats tested for co-infection by other agents, bacterial or viral? Any other testing done, histopathology etc on tissues, to determine cause of death?

Response: No, they were not tested, our strong intention was to preserve the samples exclusively for LLOV-related research. Considering the low number of carcasses we do not think practical to use these samples for other viral agents screening at this point.

6. The authors should sequence the rRNA if they cannot identify the tick to the species level by morphology. Please see the following publication that might help: <https://www.liebertpub.com/doi/10.1089/vbz.2006.6.152>

Response: Thank you for your valuable comment. We performed molecular species identification on the tick samples of the study. 33 samples were identified as Ixodes simplex and two were Ixodes vespertilionis. The mitochondrial 16S rDNA sequences were determined. We added the novel data into the manuscript text and we also uploaded these sequences to the GenBank database under the accession numbers: OL795929-OL795963

7. Oral swabs were notably absent from the sampling regimen. Any particular reason? MARV, RAVV and BOMV have all been detected in oral secretions.

Response: We understand this concern, but the animal sampling permission limited the sampling strategy and unfortunately during the research period, oral sampling was not a part of this sampling permission. We are grateful for this suggestion and we plan to include this sample type in our future sampling activities.

8. Can the authors comment on the rough size of the population at the study site and the degree to which it fluctuates over the year? Is there a resident population during winter, how big? Population in summer, how big or fold increase over winter size? Is it known roughly what percent of adult females get pregnant each year?

Response: Thank you for this valuable comment. The current knowledge and observations about the colony is summarized on Supplementary Fig 1. There are unique events in this colony, such as the large number of Rhinolophid bat arrival in the cave, which we experienced in 2021 (outside of the study period which is presented in the manuscript). The number of individuals are changing between each year from several thousand up to fifteen thousand individuals during the summer period. During winter, most of the colony leaves to a yet unknown wintering place and maximum several dozen individuals stay in the cave. Unfortunately we have data deficiency about the fluctuation of the exact colony size during the whole period of this study, therefore we were not able to use it in the discussion of our results.

Reviewers' Comments:

Reviewer #1:

Remarks to the Author:

I would like to thank the authors for making such detailed revisions and for addressing my concerns. This manuscript reads nicely and it presents data without over-exaggeration of findings. I wish the authors best of luck in their research endeavours.

Reviewer #2:

Remarks to the Author:

This is a much improved version of the manuscript. I note that the authors have added Ravn virus as another example of a filovirus that has been isolated from bats but I didn't see a reference associated with this addition. This should be corrected prior to publication.

Reviewer #3:

Remarks to the Author:

This is a very informative and well written paper that describes an exciting discovery. The revisions and additional testing, to the extent it could be done with very limited sera available, have satisfied this reviewer's concerns. Further they have appropriately modified the manuscript address the other queries. Note a very minor point at the end of the discussion and for consistency, the authors should change 'MARV and Ravn' to 'MARV and RAVV', the two abbreviations for the two marburgviruses, Marburg virus and Ravn virus.

Response to Reviewers' comments

I would like to express my gratitude for the valuable work of the reviewers. Here are the detailed responses for the comments.

Reviewer #1 (Remarks to the Author):

I would like to thank the authors for making such detailed revisions and for addressing my concerns. This manuscript reads nicely and it presents data without over-exaggeration of findings. I wish the authors best of luck in their research endeavours.

Response: Thank you for your valuable work during the revision process.

Reviewer #2 (Remarks to the Author):

This is a much improved version of the manuscript. I note that the authors have added Ravn virus as another example of a filovirus that has been isolated from bats but I didn't see a reference associated with this addition. This should be corrected prior to publication.

Response: Thank you for your valuable work during the revision process. We added the requested reference.

Reviewer #3 (Remarks to the Author):

This is a very informative and well written paper that describes an exciting discovery. The revisions and additional testing, to the extent it could be done with very limited sera available, have satisfied this reviewer's concerns. Further they have appropriately modified the manuscript address the other queries. Note a very minor point at the end of the discussion and for consistency, the authors should change 'MARV and Ravn' to 'MARV and RAVV', the two abbreviations for the two marburgviruses, Marburg virus and Ravn virus.

Response: Thank you for your valuable work during the revision process. We modified as requested.